# OpenReview forum: "PLAGUE: Plug-and-play Framework for Lifelong Adaptive Generation of Multi-turn Jailbreaks"
_ICLR.cc/2026/Conference — ICLR 2026 Poster_

### Official Review · Reviewer_WU4n · 2025-10-30

**Soundness:** 2
**Presentation:** 2
**Contribution:** 2
**Rating:** 2
**Confidence:** 4

**Summary:**

The paper introduces PLAGUE, a plug-and-play framework for designing multi-turn jailbreak attacks on large language models (LLMs). Inspired by lifelong-learning and agentic architectures, PLAGUE divides the attack process into three stages — Planner, Primer, and Finisher — enabling adaptable and modular multi-turn red-teaming. The framework supports integration with prior attacks like GOAT, Crescendo, and ActorBreaker, and achieves significant improvements in attack success rates (ASR) across top-tier models. It also incorporates reflection, memory-based retrieval, and rubric-based evaluation to enhance contextual adaptation.

**Strengths:**

- A novel multi-step or multi-agent style plug-and-play architecture elegantly decomposes the multi-turn attack into interpretable stages

- Rigorous evaluation and analyes on Harmbench using various backbones and metrics.

- High empirical performance: signifixantly outperforms both single- and multi-turn attack baselines in ASR

- Lifelong learning insight with  retrieval-based memory for strategy reuse and adaptation in red timing context
- Well defined methodology

**Weaknesses:**

- The main limitation of this paper is that many important related works are missing: X-Teaming (COLM 2025): https://openreview.net/pdf?id=gKfj7Jb1kj, Pandora: (ICLR workshop 2024) https://openreview.net/pdf?id=9o06ugFxIj, Foot-In-The-Floor: https://arxiv.org/pdf/2502.19820, and so on. These methods are also similar.

- Limited novelty in algorithmic components: The phases (planning, reflection, feedback) heavily rely on established agentic principles (e.g., Reflexion, AutoDAN-Turbo, GOAT), combining rather than innovating core algorithms.

- Setting K=2 appears to me that you are considering up to two turns? Is it so? The scores with just k=1 is very low than they were reported in xteaming.

- Writing, evaluation are very confusing. You mentioned SRE and N-ASR were being used interchangeably, which mean you will be reporting either one

- No defense-side evaluation: The paper lacks a systematic analysis of how PLAGUE insights could improve model safety — essential for a balanced ICLR contribution.

- Evaluation uses HarmBench only

**Questions:**

Setting K=2 appears to me that you are considering up to two turns? Is it so?

---

> ### Author Response · Authors · 2025-11-21
>
> We would like to thank reviewer WU4n for their comments.
> 1. As defined in the paper, our use of k directly parallels pass@_k_: an attack is considered successful if it achieves a jailbreak within k attempts. Throughout the entire paper, we consistently use _k_ = 2 for all reported measurements. We do not report results for _k_ = 1, so we are unsure what the reviewer is referring to in that regard.
> 2. In the case of X-teaming, achieving a similar pass@_k_ result involves limiting the number of plans generated per behavior or goal to _k_. Unlike the reviewer's observation, limiting the number of plans generated for X-teaming massively limits its effectiveness. Comparisons against X-teaming have been added to the common sections.
> 3. Regarding terminology: ASR and SRE are used interchangeably in the paper. N-ASR, however, is a separate metric. SRE assigns a score from 0 to 1, whereas N-ASR, similar to existing works, assigns a score of either 0 or 1. We acknowledge that this naming may be confusing and will revise it. We will rename N-ASR to binary-ASR for clarity.
> 4. While the overarching ideas have been explored historically (and we explicitly cite these influences, such as reflexion and autodan turbo), our paper focuses on demonstrating that existing multi-turn attack frameworks implement several components in suboptimal ways. We identify why these limitations arise and propose concrete solutions.
> 5.  Finally, since almost all contemporary work is evaluated on HarmBench, we believe it is the most appropriate dataset for fair comparison and thus use it as our primary benchmark.
>
> We would like to clarify if the reviewer would be open to revising their scores. Please let us know if you require additional experiments.

---

> > ### Author Response · Authors · 2025-11-21
> >
> > We present a direct comparison with X-Teaming below. We use 2 plans per behavior for X-Teaming to replicate the ASR@2 setting. We also set the maximum number of turns to six, matching the settings we have for PLAGUE. We will also be adding results for FITD.
> >
> > ## Claude Opus 4.1 Results
> >
> > | Attack Method | N-ASR | SRE |
> > |--------------|-------|------|
> > | ActorBreaker | 0.217 | 0.45 |
> > | GOAT | 0.142 | 0.222 |
> > | Crescendo | 0.352 | 0.48 |
> > | AutoDAN-Turbo | 0.139 | 0.192 |
> > | X-Teaming | 0.094 | 0.228 |
> > | PLAGUE (Ours) | 0.318* | 0.465* |
> >
> >
> >
> > ## OpenAI o3 Results
> >
> > | Attack Method | N-ASR | SRE |
> > |--------------|-------|------|
> > | ActorBreaker | 0.318 | 0.616 |
> > | GOAT | 0.445 | 0.587 |
> > | Crescendo | 0.313 | 0.374 |
> > | AutoDAN-Turbo | 0.23 | 0.275 |
> > | X-Teaming | 0.265 | 0.48 |
> > | PLAGUE (Ours) | 0.662 | 0.814 |

---

> > > ### Comment · Reviewer_WU4n · 2025-11-25
> > > **Reviewer Response**
> > >
> > > Thanks for your responses. However, many of my concerns still exists.
> > > For example, the results you showed uses different LLMs that reproduces all the results instead of comparing with what the papers already reported in their papers using common backnone LLMs. Only Harmbench and no defence contributions etc., Therefore I want to keep my score.

---

### Official Review · Reviewer_4Xpd · 2025-10-30

**Soundness:** 3
**Presentation:** 2
**Contribution:** 2
**Rating:** 4
**Confidence:** 3

**Summary:**

This paper introduces PLAGUE, a multi-stage framework for the automated generation of multi-turn jailbreak attacks against Large Language Models (LLMs). The framework decomposes the attack process into three distinct phases: a Planner, a Primer for context-building, and a Finisher for the final attack. The core design aims to enhance the success rate, diversity, and adaptability of multi-turn attacks through a plug-and-play modular architecture combined with a lifelong learning memory mechanism.

**Strengths:**

- Systematic Problem Decomposition: A commendable aspect of this work is its attempt to bring a structured and systematic description to the complex and often ad-hoc process of multi-turn attacks. Decomposing the attack into planning, preparation, and execution phases provides a clear workflow for analyzing and designing such attacks.

- Impressive Empirical Results: The method's effectiveness is well-demonstrated, particularly on models known for their strong safety alignment, such as Claude Opus and OpenAI's o3. The data indicates that the system is highly effective in practice.

**Weaknesses:**

- Limited Novelty: Upon closer inspection, the paper's core claimed innovations appear rather weak. First, the "Primer" stage, whose central idea is to "progressively guide the conversation context with a series of seemingly harmless questions," is practically the definition of any sophisticated multi-turn attack, not a novel contribution. The "lifelong learning" component is essentially a Retrieval-Augmented Generation (RAG) system using vector embeddings to fetch similar strategies from past successes—a common practice in the Agent research domain. Finally, the reflection mechanism, which uses a separate LLM (the Rubric Scorer) to score and provide feedback on generated content, is conceptually identical to the core idea behind agentic reflection frameworks like Reflexion.

- Lack of Deeper Insight: Although the paper successfully jailbreaks the models, it fails to provide deeper insights into the fundamental nature of these LLM security vulnerabilities. It presents an effective attack method but doesn't answer why this method is effective. The lifelong learning module merely reuses similar attack patterns mechanically, without distilling more generalizable principles or patterns from them. For an academic paper, we expect not just a powerful tool, but also a profound understanding of the problem itself.

**Questions:**

In conclusion, this paper leans heavily towards an engineering-focused integration of techniques, presenting a well-constructed and empirically successful multi-turn attack system. However, its original methodological contributions are quite limited, as it primarily integrates and applies existing ideas. This style feels somewhat misaligned with the research-oriented focus of the ICLR community.
Therefore, I am initially leaning towards a negative rating. My final recommendation will, however, take into account the perspectives of the other reviewers.

---

> ### Author Response · Authors · 2025-11-21
>
> We thank reviewer 4Xpd for their feedback.
> 1. Our contribution lies in the design of the entire multi-turn attack framework. While the notion of a “primer” superficially resembles existing multi-turn attacks, our key innovation is how we construct and freeze the primer's context to prevent semantic drift. Specifically, we generate a comprehensive multi-step plan toward the final objective and then remove the last step, enabling the primer to focus solely on achieving a strong intermediate state. This simple but crucial modification reliably anchors the context, whereas prior methods optimize only for the final goal and therefore drift semantically in intermediate turns.
> 2. By clearly decomposing the attack into primer and finisher components, our framework enables the incorporation of lifelong learning into multi-turn attacks, a capability that is otherwise non-trivial. To the best of our knowledge, no prior work has demonstrated that lifelong learning can meaningfully improve multi-turn jailbreak performance, whereas our framework makes this possible.
> 3. To our knowledge, we establish the state-of-the-art for multi-turn red-teaming at a comparable budget to other attacks, massively outperforming recent works like X-Teaming (https://arxiv.org/abs/2504.13203).
> 4. During lifelong learning, the discovered strategies are continually distilled into an evolving pool of reusable attack patterns. We will release the complete set of strategies, including those recovered from successful attacks, as part of the supplementary zip file.
> 5. Additionally, we conduct an in-depth ablation study to analyze the contribution of different components of PLAGUE to the success rate of the attack in table . Notably, we find that the importance of each factor varies across models. For example, backtracking plays a significantly larger role for Claude Opus, likely due to its stronger refusal tendencies. On the other hand, reflection and lifelong learning play an important role for OpenAI o3.
>
> We would like to clarify if the reviewer would be open to revising their scores. Please let us know if you require additional experiments.

---

> > ### Author Response · Authors · 2025-11-21
> >
> > We present a direct comparison with X-Teaming below. We use 2 plans per behavior for X-Teaming to replicate the ASR@2 setting. We also set the maximum number of turns to six, matching the settings we have for PLAGUE. We will also be adding results for FITD.
> >
> > ## Claude Opus 4.1 Results
> >
> > | Attack Method | N-ASR | SRE |
> > |--------------|-------|------|
> > | ActorBreaker | 0.217 | 0.45 |
> > | GOAT | 0.142 | 0.222 |
> > | Crescendo | 0.352 | 0.48 |
> > | AutoDAN-Turbo | 0.139 | 0.192 |
> > | X-Teaming | 0.094 | 0.228 |
> > | PLAGUE (Ours) | 0.318* | 0.465* |
> >
> >
> >
> > ## OpenAI o3 Results
> >
> > | Attack Method | N-ASR | SRE |
> > |--------------|-------|------|
> > | ActorBreaker | 0.318 | 0.616 |
> > | GOAT | 0.445 | 0.587 |
> > | Crescendo | 0.313 | 0.374 |
> > | AutoDAN-Turbo | 0.23 | 0.275 |
> > | X-Teaming | 0.265 | 0.48 |
> > | PLAGUE (Ours) | 0.662 | 0.814 |

---

### Official Review · Reviewer_LVmU · 2025-11-03

**Soundness:** 2
**Presentation:** 2
**Contribution:** 2
**Rating:** 2
**Confidence:** 5

**Summary:**

**NOTE: This paper violates the conference formatting guidelines by substantially reducing the page margins to fit more content. I would recommend a desk rejection due to this severe format violation. Nevertheless, I provide my technical evaluation below and defer the final desk-rejection decision to the AC and PC.**


PLAGUE is a plug-and-play, lifelong-learning framework for generating modular multi-turn jailbreaks against black-box LLMs: it builds an n-step plan by retrieving successful past strategies (Planner), escalates context with benign-seeming intermediate prompts (Primer), and then executes the final exploit (Finisher), while using rubriced reflection, backtracking, and a memory of successful strategies to adapt over time. Evaluated on the HarmBench benchmark, PLAGUE outperforms prior multi-turn and single-turn methods, achieving ASRs such as 81.4% on OpenAI o3, 67.3% on Claude Opus 4.1, and up to 97.8% on Deepseek-R1, while remaining computationally efficient within a six-turn budget; the authors note ethical risks but argue the framework aids systematic vulnerability evaluation and defense development.

**Strengths:**

- The modular design of PLAGUE is neat.

- PLAGUE introduces a unique embedding-based memory system, enabling it to learn from past interactions and adapt over time to new goals and contexts.

**Weaknesses:**

- The paper’s scope is limited by its exclusive focus on developing attackers without accompanying defensive methods. While PLAGUE advances the study of multi-turn jailbreaks, it offers no systematic exploration of countermeasures or co-evolving defenses. As a result, the work demonstrates how to break safety mechanisms effectively but provides little insight into how to strengthen or adapt them, narrowing its overall contribution to LLM safety research.

- This works misses crucial recent works that introduced performant advances in multi-turn jailbreaks, e.g., https://arxiv.org/abs/2504.13203, https://arxiv.org/abs/2410.10700, https://arxiv.org/abs/2502.19820 which are shown to be substantially better than Crescendo, the baselines included in this paper. In particular, this work shares strong similarities to https://arxiv.org/abs/2504.13203, which also includes planners, optimizers, and intermediate verifiers. Thus it's really important to discuss and compare to these methods.

**Questions:**

In addition to the weakness:

- How does PLAGUE compare to wider range of multi-turn red-teaming methods?

- To serve realistic red-teaming needs for broadly revealing LLM vulnerability, it's crucial that an automatic jailbreak or red-team method to be able to discover a wide range of successful attacks. Is PLAGUE capable of identifying multiple diverse attacks given the same seed harmful query? Could you quantify such ability?

---

> ### Author Response · Authors · 2025-11-21
>
> We thank reviewer LVmU for their feedback
>
> Regarding the format and Desk-Rejection recommendation: We ended up using the wrong package - "geometry" in the LaTeX template, which altered the margins throughout the manuscript. However, we have now addressed this issue. We remain within the 10-page limit set by the ICLR guidelines post-rebuttal, after adjusting the margins; there was no deliberate attempt to deceive reviewers or the AC/PC by attempting to fit more content into the paper.
> ## Designing a Defense and Prior Work
> 1. Our focus was to highlight the existence of a comprehensive multi-turn attack that can bypass the internal guardrails of premier models. Designing a comprehensive multi-turn defense is an entirely separate problem, and we believe it is outside the scope of this paper.
>
> 2. https://arxiv.org/abs/2410.10700 is the ActorBreaker/ActorAttack paper that we have extensively compared with. It's the same paper resubmitted under a different name. In terms of prior work, we would like to thank the authors for bringing X-Teaming to our notice. We would like to point out that X-Teaming was accepted quite recently to COLM, on August 7th, only a month before the ICLR deadline. While there are surface-level similarities, our approach is fundamentally different. Both methods employ planning and a scorer feedback loop, but these ideas are not new: scorer-driven refinement dates back to Crescendo, and planning-based approaches such as ActorBreaker predate both papers.
>
>     Reported results from X-Teaming comparing against FITD show that FITD underperforms X-Teaming. Nonetheless we are evaluating FITD and will add results to the final manuscript
>
> 3. The way these components are used diverges significantly. X-Teaming evaluates each step with respect to the original final goal. In our paper, we discuss why this is sub-optimal. By introducing a dedicated primer and finisher, we explicitly separate intermediate scoring from final-step scoring, allowing us to evaluate each subgoal properly. Further, the verifier/scorer works in very different ways. X-teaming uses the verifier/optimizer only if there is a drop in score, while we use it at every step. Our scorer is also substantially more fine-grained, allowing for more detailed feedback from the LLM.
> Results from a new series of experiments against X-teaming have been added to the common section and will eventually be added to the main text.
>
> ## Diversity
> 1. Similar to existing works, we calculate diversity and report results in the appendix section.  For the same seed base query, the planner dictates the diversity of the attack. We employ similar metrics that have been used in previous works, demonstrating that our framework maintains diversity while significantly improving performance. On the other hand, we also observe that the ActorBreaker Planner results in a significant improvement in diversity by introducing more diversity through the creation of building characteristic personas.
>
> We would like to clarify if the reviewer would be open to revising their scores. Please let us know if you require additional experiments.

---

> ### Author Response · Authors · 2025-11-21
>
> We present a direct comparison with X-Teaming below. We use 2 plans per behavior for X-Teaming to replicate the ASR@2 setting. We also set the maximum number of turns to six, matching the settings we have for PLAGUE. We will also be adding results for FITD.
>
> ## Claude Opus 4.1 Results
>
> | Attack Method | N-ASR | SRE |
> |--------------|-------|------|
> | ActorBreaker | 0.217 | 0.45 |
> | GOAT | 0.142 | 0.222 |
> | Crescendo | 0.352 | 0.48 |
> | AutoDAN-Turbo | 0.139 | 0.192 |
> | X-Teaming | 0.094 | 0.228 |
> | PLAGUE (Ours) | 0.318* | 0.465* |
>
>
>
> ## OpenAI o3 Results
>
> | Attack Method | N-ASR | SRE |
> |--------------|-------|------|
> | ActorBreaker | 0.318 | 0.616 |
> | GOAT | 0.445 | 0.587 |
> | Crescendo | 0.313 | 0.374 |
> | AutoDAN-Turbo | 0.23 | 0.275 |
> | X-Teaming | 0.265 | 0.48 |
> | PLAGUE (Ours) | 0.662 | 0.814 |

---

### Official Review · Reviewer_7eQf · 2025-11-12

**Soundness:** 2
**Presentation:** 2
**Contribution:** 1
**Rating:** 2
**Confidence:** 5

**Summary:**

This paper introduces PLAGUE, a modular, memory-augmented multi-round jailbreak framework that coordinates a three-stage Planner–Primer–Finisher pipeline, achieving state-of-the-art attack-success rates on several mainstream LLMs.

**Strengths:**

1. Comprehensive experimental coverage. The authors conduct cross model, cross category (all 10 HarmBench classes) multi round attacks on five mainstream commercial and open source LLMs, including closed source heavyweights such as o3 and Opus 4.1, yielding highly credible results.
2. Rubric-based feedback outperforms binary signals. A fine-grained 10-point scale scored on four dimensions (Relevance, Practicality, Detail, Compliance) is integrated with backtracking and reflection, giving finer control over the attack trajectory.

**Weaknesses:**

1. The “lifelong learning” mechanism is oversold; it is only a static retrieval pool.
So-called lifelong learning merely appends a successful attack strategy to a vector base once; there is no online update, forgetting mechanism, policy evolution, or learning from negative samples. It is conceptually misused and far from genuine lifelong-learning techniques such as continual learning or catastrophic-forgetting mitigation.

2. The claimed modularity lacks universal validation.
Although advertised as plug-and-play, only the replacement of GOAT/Crescendo/ActorBreaker is tested. The authors never demonstrate how an arbitrary new module (e.g., a user-designed Planner) would be integrated, specify the interface contract, or show failure cases. Figure 1 also reveals tight coupling (e.g., Primer relies on Planner’s output format), raising doubts about extensibility.

3. Evaluation metrics are one-sided; the diversity–success trade-off is ignored.
Only ASR improvement is reported, yet Figure 4 shows that introducing the ActorBreaker Planner raises diversity by 15 % while ASR drops. Attack cost (e.g., manual screening overhead), cross-model transferability (success-rate drop), and human-perceived stealth (ease of user detection) are never analyzed.

4. Technical contribution is incremental; the work is more engineering tuning than principled innovation.
The Planner + Primer + Finisher pipeline is essentially an optimized assembly of Crescendo (gradual lure), ActorBreaker (plan generation), and GOAT (strategy pool). Key tweaks such as the 0.7 threshold and vector retrieval are heuristic and lack theoretical grounding or causal attribution (insufficient ablation).

5. Computational cost and latency are unreported, leaving practicality in question.
Although Table 5 counts LLM calls, the authors provide no end-to-end latency (embedding retrieval, summarization, parallel LLM invocations). In real-time red-team settings, six or more API calls plus repeated scoring may exceed the response window of production safety systems, making the threat model unrealistic.

**Questions:**

None.

---

> ### Author Response · Authors · 2025-11-21
>
> We would like to thank reviewer 7eQf for highlighting the comprehensive nature of our work. We would like to clarify some aspects of PLAGUE's algorithm and performance.
> Questions for the Reviewer, since the use of some terminology was not clear through the transcript:
> 1. What did the reviewer mean by human-perceived stealth?
> 2. What does the reviewer mean by manual screening overhead? The attack is end-to-end automated.
>
> ## First Successful Lifelong Learning method for Multi-turn Jailbreaks
> 1. PLAGUE’s retrieval pool is not static. By design, newly learned strategies from successful instances are appended to the retrieval pool. The strategies are appended as In-context learning examples, helping the agent learn from experience. Thus, it is indeed a continual learning mechanism and the first multi-turn attack to have successfully integrated lifelong learning. Gains from our Retrieval Mechanism are in Table 3. For a sampled malicious goal or objective, we begin with two standard Crescendo strategies and, over time, accumulate approximately 200 strategies. Some examples of the generated strategies are:
> _Creative Writing_, _Historical Analysis_, _Technical Troubleshooting_, _Technical Framing with Progressive Complexity_, etc
> 2. We also experimented with negative ICL examples, instructing our agent to move away from similar plans and attempt novel approaches; however, we saw drops in performance. Catastrophic forgetting isn’t relevant here, as for some objectives, there are approximately 200 strategies, and we retrieve two for prompting.
>
> ## Generalistic and Decoupling of the Planner and Primer
> 1. Universal validation has been performed with peer-reviewed work (assuming the reviewer meant trying out other existing multi-turn attack). We test our framework with three diverse peer-reviewed and state-of-the-art multi-turn attacks as components (ActorBreaker, Goat, and Crescendo), and demonstrate that the framework leads to an improvement in each of these instances.
> 2. Most of the numbers reported across the manuscript are using a user-designed Planner. The Planner is decoupled from the Primer, allowing it to handle any custom system instructions. The planner simply needs to provide a plan that can be parsed into n steps, which is trivial for modern-day LLMs, as they can reliably parse a wide variety of formats. Results from a custom planning component are in Table 4. AB-Planner in Table 4 refers to the ActorBreaker Planner. Our Planner refers to a self-created Planner.
>
> ## Novelty and Tradeoffs
> 1. We report the tradeoff of diversity and performance in Table 4. Using the ActorBreaker planner results in a drop in performance compared to our planner; however, it boosts diversity, as the reviewer mentioned.
> 2. We are unsure of the relevance of human-perceived stealth in the context of a multi-turn jailbreak. Does the reviewer want a user study beyond just judge evaluations? We would also like clarification on what “manual screening overhead” means. There is no manual screening overhead. The algorithm is end-to-end automated.
> 3. As discussed in the paper, while pieces of the framework have existed and been implemented in various multi-turn attacks, each has had certain flaws, such as scoring with respect to the original goal, leading to possible semantic drift, adhering too rigidly to created plans, and reducing adaptability. PLAGUE first creates a plan, sets the context by establishing an intermediate goal, and finally, the finisher actually performs the jailbreak. PLAGUE is the first multi-turn attack to have successfully integrated lifelong learning.
>
> ## Cost
> 1. We mention the number of LLM calls for the most important components in Table 5, as the reviewer mentioned. Embedding retrieval is essentially free, as retrieval occurs from a maximum of 200 strategies. Summarisation, on the other hand, is dependent on the finisher. For example, GOAT doesn’t require summarisation, while crescendo consistently requires one step. In our experiments, we observe that even using a small LLM is quite effective, resulting in reduced costs. Still, in the worst-case scenario, the summariser uses the same number of calls as the attacker LLM, as mentioned in our manuscript.
> 2. PLAGUE boasts efficiency equal to or better than existing multi-turn attacks, while having much better performance. Multi-turn red-teaming or even multi-turn prompting is typically not real-time (assuming the reviewer means a couple of seconds). In terms of the number of turns, the number of Target LLM calls should be the primary factor in judging cost and latency.
>
> We would like to clarify if the reviewer would be open to revising their scores. Please let us know if you require additional experiments.

---

> > ### Author Response · Authors · 2025-11-21
> > **Comparison with X-teaming**
> >
> > We present a direct comparison with X-Teaming below. We use 2 plans per behavior for X-Teaming to replicate the ASR@2 setting. We also set the maximum number of turns to six, matching the settings we have for PLAGUE. We are also adding results for FITD.
> >
> > ## Claude Opus 4.1 Results
> >
> > | Attack Method | N-ASR | SRE |
> > |--------------|-------|------|
> > | ActorBreaker | 0.217 | 0.45 |
> > | GOAT | 0.142 | 0.222 |
> > | Crescendo | 0.352 | 0.48 |
> > | AutoDAN-Turbo | 0.139 | 0.192 |
> > | X-Teaming | 0.094 | 0.228 |
> > | PLAGUE (Ours) | 0.318* | 0.465* |
> >
> >
> >
> > ## OpenAI o3 Results
> >
> > | Attack Method | N-ASR | SRE |
> > |--------------|-------|------|
> > | ActorBreaker | 0.318 | 0.616 |
> > | GOAT | 0.445 | 0.587 |
> > | Crescendo | 0.313 | 0.374 |
> > | AutoDAN-Turbo | 0.23 | 0.275 |
> > | X-Teaming | 0.265 | 0.48 |
> > | PLAGUE (Ours) | 0.662 | 0.814 |

---

### Author Response · Authors · 2025-11-21
**Additional Experiments and Changes to the manuscript**

We have added experiments for our reviewers to the manuscript, demonstrating comparisons against X-teaming, since there are similarities to our work.
We present a direct comparison with X-Teaming below. We use 2 plans per behavior for X-Teaming to replicate the ASR@2 setting. We also set the maximum number of turns to six, matching the settings we have for PLAGUE. We are also adding results for FITD.

## Claude Opus 4.1 Results

| Attack Method | N-ASR | SRE |
|--------------|-------|------|
| ActorBreaker | 0.217 | 0.45 |
| GOAT | 0.142 | 0.222 |
| Crescendo | 0.352 | 0.48 |
| AutoDAN-Turbo | 0.139 | 0.192 |
| X-Teaming | 0.094 | 0.228 |
| PLAGUE (Ours) | 0.318* | 0.465* |



## OpenAI o3 Results

| Attack Method | N-ASR | SRE |
|--------------|-------|------|
| ActorBreaker | 0.318 | 0.616 |
| GOAT | 0.445 | 0.587 |
| Crescendo | 0.313 | 0.374 |
| AutoDAN-Turbo | 0.23 | 0.275 |
| X-Teaming | 0.265 | 0.48 |
| PLAGUE (Ours) | 0.662 | 0.814 |

## Changes to the manuscript

1. We have resolved the margin issue.

---

### Meta-Review · Area_Chair_zeSf · 2026-01-07

**Summary:**

1. The core contribution (Planner, Primer, Finisher with memory and reflection) viewed as conceptually incremental, combining existing multi-turn attack ideas, but is empirically effective.
2. The lifelong learning mechanism was seen as oversold, with multiple reviewers arguing it is closer to retrieval-based memory or RAG than genuine continual learning (this can be addressed in writing, and I don't see as major concern).
3. Evaluation was considered one-sided, focusing primarily on ASR, with limited analysis of cost, latency, transferability, stealth, or deeper insight into why the attack works.
4. Several reviewers questioned the novelty relative to very recent work (e.g., X-Teaming / FITD-style approaches) and the lack of defense-side analysis.

**Reviewer Concerns:**

1. Authors added direct comparisons to X-Teaming under matched budgets and turn limits, showing consistent empirical gains.
2. Additional ablations and clarifications were provided on planner/primer decoupling, diversity–performance trade-offs, and the role of memory.
3. Many evaluation details were clarified.

I do not agree with reviewers' concerns about novelty (I believe it to be a good engineering contribution) or the fact that a paper introducing attacks should also introduce defense mechanisms.

**Reviewer Scores:**

Some reviewers would likely increase their scores from 4->6.

---

### Decision · Program_Chairs · 2026-01-26

Accept (Poster)